# Ebselen and Diphenyl Diselenide Inhibit SARS-CoV-2 Replication at Non-Toxic Concentrations to Human Cell Lines

**DOI:** 10.3390/vaccines11071222

**Published:** 2023-07-10

**Authors:** Guilherme Wildner, Amanda Resende Tucci, Alessandro de Souza Prestes, Talise Muller, Alice dos Santos Rosa, Nathalia Roberto R. Borba, Vivian Neuza Ferreira, João Batista Teixeira Rocha, Milene Dias Miranda, Nilda Vargas Barbosa

**Affiliations:** 1Programa de Pós-Graduação em Bioquímica Toxicológica, Universidade Federal de Santa Maria, Santa Maria 97105-900, RS, Brazil; wildner.gw@gmail.com (G.W.); prestes.asp@gmail.com (A.d.S.P.); talise_tm@yahoo.com.br (T.M.); joao.rocha@ufsm.br (J.B.T.R.); 2Laboratório de Morfologia e Morfogênese Viral, Instituto Oswaldo Cruz, Fundação Oswaldo Cruz, Rio de Janeiro 21041-250, RJ, Brazil; amanda.tucci@ioc.fiocruz.br (A.R.T.); alicerosa@aluno.fiocruz.br (A.d.S.R.); nathaliaborbahr18@gmail.com (N.R.R.B.); vnsantos@bioqmed.ufrj.br (V.N.F.); 3Programa de Pós-Graduação em Biologia Celular e Molecular, Instituto Oswaldo Cruz, Fundação Oswaldo Cruz, Rio de Janeiro 21041-250, RJ, Brazil

**Keywords:** organoselenium molecules, ebselen, diphenyl diselenide, anti-SARS-CoV-2 activity, lymphocytes, antiviral

## Abstract

The novel severe acute respiratory syndrome coronavirus 2 (SARS-CoV-2) was the causative agent of the COVID-19 pandemic, a global public health problem. Despite the numerous studies for drug repurposing, there are only two FDA-approved antiviral agents (Remdesivir and Nirmatrelvir) for non-hospitalized patients with mild-to-moderate COVID-19 symptoms. Consequently, it is pivotal to search for new molecules with anti-SARS-CoV-2 activity and to study their effects in the human immune system. Ebselen (Eb) is an organoselenium compound that is safe for humans and has antioxidant, anti-inflammatory, and antimicrobial properties. Diphenyl diselenide ((PhSe)_2_) shares several pharmacological properties with Eb and is of low toxicity to mammals. Herein, we investigated Eb and (PhSe)_2_ anti-SARS-CoV-2 activity in a human pneumocytes cell model (Calu-3) and analyzed their toxic effects on human peripheral blood mononuclear cells (PBMCs). Both compounds significantly inhibited the SARS-CoV-2 replication in Calu-3 cells. The EC_50_ values for Eb and (PhSe)_2_ after 24 h post-infection (hpi) were 3.8 µM and 3.9 µM, respectively, and after 48 hpi were 2.6 µM and 3.4 µM. These concentrations are safe for non-infected cells, since the CC_50_ values found for Eb and (PhSe)_2_ on Calu-3 were greater than 200 µM. Importantly, the concentration rates tested on viral replication were not toxic to human PBMCs. Therefore, our findings reinforce the efficacy of Eb and demonstrate (PhSe)_2_ as a new candidate to be tested in future trials against SARS-CoV-2 infection/inflammation conditions.

## 1. Introduction

The first identified cases of infection by the novel severe acute respiratory syndrome coronavirus 2 (SARS-CoV-2) were detected in December 2019 in Wuhan, China [1], and soon after, the virus was identified as the causative agent of coronavirus disease 2019 (COVID-19). In March 2020, COVID-19 was declared a pandemic by the World Health Organization, spreading rapidly across the world and becoming one of the worst and most deadly pandemics in the world [2]. Currently, it is estimated that more than 767 million cases of infection have been recorded since the beginning of the pandemic, and that more than 6.9 million deaths have occurred worldwide [3].

SARS-CoV-2 is a β-coronavirus, composed of a capsid containing a single-strand RNA (ssRNA) and four structural proteins: the nucleocapsid protein (N), the membrane protein (M), the envelope protein (E), and the spike protein (S) [4]. The virion envelope is made up of the proteins S, M, and E, where the S protein is responsible for the entry of the virus into the host cells. The S protein attaches to the host cells through the interaction with a specific cellular receptor, angiotensin-converting enzyme 2 (ACE2), together with a host factor, the surface serine protease (TMPRSS2) [5].

Upon entry into the host cells, the viral replication and transcription complex is formed with non-structural proteins (nsps), as a result of processing two polyproteins (pp1a and pp1ab) translated from the virus genomic RNA. The main protease (Mpro, or 3C-like protease—3CLpro) and papain-like protease (PLpro) are critical sulfhydryl enzymes for the viral replication and transcription due to their post-translational action on pp1a and pp1ab processing [6]. Together with the S protein, the SARS-CoV-2 proteases became important targets of antiviral approaches and research on the mechanisms underlying the infection [7,8,9].

After continuous multiplication, the virus reaches the lungs, where it causes alveoli inflammation, leading to pneumonia. This physiological state can rapidly progress to acute respiratory distress syndrome (ARDS) [10,11]. In this phase, the hyperactivation of immune cells and excessive cytokine production are responsible for the severe inflammatory process characteristic of COVID-19, known as cytokine storm [12]. So far, the number of new SARS-CoV-2 infections and deaths from COVID-19 pathogenesis highlight the importance of new therapies, since those that currently exist are limited [13]. The U.S. Food and Drug Administration (FDA) approved the Emergency Use Authorization (EUA) for some drugs to treat mild-to-moderate COVID-19 patients likely to progress to a more severe illness. Although Remdesivir and Paxlovid (nirmatrelvir and ritonavir) are the only antiviral drugs approved by the FDA for the treatment of adults with mild-to-moderate COVID-19 who are at high risk for severe disease progression, which can lead to hospitalization and death, the antiviral Lagevrio (Molnupiravir) has also been used as a treatment under an Emergency Use Authorization, as proposed by the National Institutes of Health (NIH) [14]. However, there is still an urgent need to repurpose medication against the SARS-CoV-2 infection and acute inflammatory response of COVID-19.

Drug repurposing, molecular simulations in search of viral protease inhibitors, and screenings for antiviral/anti-inflammatory molecules have been strategies widely used in the global research against the SARS-CoV-2 virus and the effects of COVID-19. In this context, here, we highlight the study of two organic selenium compounds, ebselen (Eb) and diphenyl diselenide ((PhSe)_2_). In addition to the well-known antioxidant and anti-inflammatory effects in several experimental models of human pathologies, the compounds have been reported as antimicrobial agents by oxidizing Cys residues from critical enzymes in different microorganisms [15,16]. Importantly, one of the first computational experimental screenings toward SARS-CoV-2 identified Eb as one of the most promising inhibitors of the protease Mpro, as well as a potent inhibitor of the enzyme in vitro and of SARS-CoV-2 replication [17,18]. The antiviral activity of (PhSe)_2_ is less explored, but there is evidence of its efficacy against other viruses, including the herpes simplex virus 2 (HSV-2) and the bovine alphaherpesvirus 2 (BoHV-2) [19,20,21]. Moreover, our group has already demonstrated that (PhSe)_2_ is capable of covalently interacting with the SARS-CoV-2 Mpro in silico and inhibiting virus replication in a Vero E6 cell model.

Herein, we investigated the effects of Eb and (PhSe)_2_ on SARS-CoV-2 replication using Calu-3 cells as the virus-targeted cells. To ensure the safety of the antiviral concentrations tested, we also exposed human PBMCs to the Eb and (PhSe)_2_ to ascertain some endpoints of toxicity. We found that both compounds significantly inhibited the replication of SARS-CoV-2 in human pneumocytes, presenting low EC_50_ values, comparable to others already described in the literature for anti-SARS-CoV-2 molecules [22]. Of toxicological significance, the compounds did not induce any signs of toxicity in human PBMCs, as evaluated by parameters of viability loss, apoptosis, morphological changes, and cell cycle disruption. In the peroxidation assay, the compounds also did not exhibit pro-oxidant action at any concentration tested.

## 2. Materials and Methods

### 2.1. Eb and (PhSe)_2_ Anti-SARS-CoV-2 Activity in Calu-3 Cells

Calu-3 (cell line from the submucosal gland that recapitulates type II pneumocytes) cells were infected with a SARS-CoV-2 isolate (lineage B.1, GenBank #MT710714, SisGen AC58AE2) at a multiplicity of infection (MOI) of 0.01 or 0.1 for 1 h at 37 °C. After that, the culture medium was removed and the treatments with Eb or (PhSe)_2_, ranging from 0.78 to 12.5 µM, were performed for 24 h and 48 h. The replicative ability of SARS-CoV-2, in the presence and absence of the compounds, was evaluated by counting the plaque-forming units (PFU/mL). For the PFU assay, Vero E6 (African green monkey kidney, ATCC CRL-1586) cells, previously seeded in 96-well plates containing 2 × 10^4^ cells/well, were incubated with a serial dilution (1:200 to 1:12,800) of supernatants. After 1 h of infection, 10× Dulbecco’s Modified Eagle Medium (DMEM, Gibco, Grand Island, NY, USA) containing 2% fetal bovine serum (FBS; Gibco^TM^, Fisher Scientific, Loughborough, UK) and 2.4% carboxymethylcellulose (Sigma-Aldrich, Burlington, MA, USA) was added to the cells and the incubation was maintained at 37 °C for 72 h. The fixation of the biological material was performed with 4% formalin solution for 3 h. After that, the cells were stained with 0.04% crystal violet solution for 1 h at room temperature. Then, the material was washed in water, and after drying the quantification of the PFU was performed. All procedures involving the handling of SARS-CoV-2 were performed in a biosafety level 3 (BSL3) environment, in accordance with the World Health Organization guidelines [23].

### 2.2. Human Subjects and Blood Collection

Peripheral blood was collected from healthy volunteers (4 men, 4 women, mean age of 30 ± 10). Exclusion criteria included alcohol abuse or drug dependence, diseases, and drug treatments in the last 2 weeks. The blood collection was performed following the Declaration of Helsinki. Volunteers were not exposed to any risk while participating in the study. The protocol was approved by the Ethics Committee for Research with Humans at the Federal University of Santa Maria (number 67825122.1.0000.5346).

### 2.3. Peripheral Blood Mononuclear Cells’ (PBMCs) Isolation and Treatments

Peripheral blood mononuclear cells (PBMCs) were isolated following the methodology described by Ecker et al. [24], with some modifications. In this step, peripheral blood (20 mL) was collected from each volunteer under aseptic conditions into tubes coated with sodium heparin (Hipolabor Farmacêutica Ltda., Belo Horizonte, MG, Brazil). Whole blood was diluted 1:1 with PBS buffer (136 mM NaCL, 2.68 mM KCL, 1.47 mM KH_2_PO_4_, 8.1 mM Na_2_HPO_4_, pH 7.4) and layered over Ficoll-Paque Plus (GE Healthcare, Piscataway, NJ, USA) before the centrifugation at 577× *g* for 30 min to separate PBMCs. Cells were washed three times with PBS (followed by centrifugations at 400× *g* for 10 min, 256× *g* for 10 min, and 178× *g* for 5 min, respectively), and then cultured in RPMI media supplemented with 2% PHA, 10% BFS, and 1% antibiotic–antimycotic, containing Eb and (PhSe)_2_ (both diluted in DMSO p.a as a vehicle). PBMCs were cultured in ELISA microplates maintained in culture for 24 h at 37 °C and 5% CO_2_. The number of cells used was 0.5 × 10^6^ PBMCs per group. Two controls were used (with and without final DMSO of 0.25%). However, only the PBS buffer control is shown in the results since there was no statistical difference between these groups among all the evaluated parameters.

### 2.4. Cytotoxicity Assays

The possible toxicity of compounds in the cell models was determined by the MTT (3-(4,5-dimethyl-2-thiazolyl)-2,5-diphenyl-2H-tetrazolium bromide; Sigma-Aldrich) assay following the methodology described by Mosmann [25], with some modifications. Herein, PBMCs and Calu-3 were exposed to different concentration curves, ranging from 0.25 to 200 µM in PBMCs or 25 to 200 µM in Calu-3, of the tested compounds (dissolved in DMSO p.a, final: 0.1% *v*/*v*) for 24 or 72 h, respectively. After the treatment period, 0.16 mg/mL or 5 mg/mL of MTT, previously diluted in PBS, was added to the medium of PBMCs or Calu-3, respectively. Then, the samples were incubated for 2 h at 37 °C, in the dark. Subsequently, the samples were centrifuged, the supernatant was discarded, and the formazan crystals were solubilized with 10% SDS or DMSO p.a. After 2 h, the samples were read in an ELISA microplate reader at 540 or 570 nm (TP-Reader Thermo Plate), depending on the cell type, and the dehydrogenase enzyme activity was expressed as a percentage of the control, only exposed to DMSO at the same concentration used in the treated conditions.

### 2.5. Lipid Peroxidation by the Thiobarbituric Acid-Reactive Substances (TBARS) Method

Lipid peroxidation was assessed to evaluate the possible pro-oxidant activity of the compounds. The capacity of the compounds to induce oxidation of the glycerophospholipid phosphatidylcholine was tested by the TBARS method, as described by Ohkawa [26] and Prestes [27], with minor modifications. Briefly, the phosphatidylcholine (4.29 mg/mL) was incubated for 30 min at 37 °C with 10 mM of Tris buffer, pH 7.4, and fresh Fe_2_SO_4_ (214 µM, pro-oxidant control), in the absence or presence of (PhSe)_2_ or Eb (0.5, 1, 2, 5, 10, 25 µM). The reaction was stopped with SDS (final 1.35%). The color reaction was developed by incubating the medium with acetic acid/HCL buffer, pH 3.4, and 0.2% TBA, pH 6, for 60 min at 100 °C. The levels of TBARS were measured at 532 nm. The results were expressed as % of the control.

### 2.6. Morphological Parameters

Morphological changes of PBMCs (5 × 10^5^ cells/sample) were analyzed after 24 h of exposure to (PhSe)_2_ or Eb at 2.5 µM in a flow cytometry apparatus (BD Accury C6 Cytometer). In this test, the size and granularity were verified through digital signals of forward scatter (FSC) and side scatter (SSC), respectively. A total of 100,000 events were acquired for each sample, as described by Prestes et al. [28], with adaptations.

### 2.7. Viability, Apoptosis, and Necrosis Indexes

Cell viability, early apoptosis, advanced apoptosis, and necrosis rates were measured by the propidium iodide (PI) plus Annexin V kit (Alexa Fluor^®^ 488 Annexin V/Dead Cell Apoptosis Kit). After 24 h of exposure to (PhSe)_2_ or Eb at 2.5 µM, PBMC samples (5 × 10^5^ cells/sample) were processed as indicated by the manufacturer, and analyzed in a flow cytometer (BD Accury C6 Cytometer), where the rate of viable cells (not marked by Annexin V and PI), cells in early apoptosis (marked only by Annexin V), cells in advanced apoptosis (marked by both Annexin V and PI), or necrotic cells (marked only by PI) were quantified. To avoid signal overlap, fluorescence compensation was performed for the FL1 (Annexin) and FL3 channels (PI). Here, 100,000 events were recorded per sample, as described by Ecker et al. [29].

### 2.8. Cell Cycle Assay

The phases of the cell cycle were analyzed according to William-Faltaos et al. [29] and Ecker et al. [29], with minor modifications. Here, 0.5 × 10^6^ cells/mL per group of PBMCs were seeded into 96-well plates (200 µL per sample) containing (PhSe)_2_ or Eb at 2.5 µM for 24 h. Then, the cells were removed, washed with PBS buffer, and fixed in ethanol (70%) for 24 h, at −20 °C. Subsequently, PBMCs were centrifuged (500× *g* for 10 min), washed once with PBS, and then resuspended in a labeling solution (30 mM PI in PBS buffer containing 200 mg/mL Dnase-free Rnase and 0.1% Triton-X) during 30 min in the dark. Then, the G0–G1, S, and G2/M phases were analyzed by flow cytometry using the FL2 channel. Here, 100,000 events were acquired for each sample.

### 2.9. Statistical Analysis

One-way ANOVA followed by Tukey’s or Dunnett’s multiple tests, as appropriate, were used to perform statistical analyses. Results were expressed as mean ± SD or mean ± SEM. All experiments were independently carried out three to six times. Differences were considered statistically significant when *p* ≤ 0.05. The GraphPad Prism version 8.00 for Windows (GraphPad Software, La Jolla, CA, USA) was used to create the graphics.

## 3. Results

### 3.1. MTT and TBARS Assays

The performance of diphenyl diselenide ((PhSe)_2_) and ebselen (Eb) on the viability of distinct cellular lines was initially evaluated in this study. For a better understanding and a more satisfactory discussion, the chemical structures of these molecules are presented in Figure 1.

Via the MTT assay, we found that both (PhSe)_2_ and Eb induced a significant decrease in the PBMCs’ dehydrogenase activity from 50 µM. From the concentration curves, the IC_50_ values calculated for (PhSe)_2_ and Eb were seen to be 38.75 and 34.84 µM, respectively (Figure 2).

Toxicity was also evaluated in the SARS-CoV-2 cellular infection model used in this study. Thereby, the viability of non-infected cells was ensured in treatments with high concentrations of selenium compounds (25–200 µM), and the percentage of viable cells of each treatment was obtained in comparison to the control samples treated only with DMSO, at the same percentages as those to which the ebselen- and diphenyl diselenide-treated cells were subjected (Figure 3). We found that both compounds were less toxic in Calu-3 than in PBMCs, which was reflected in the CC_50_ of 200 µM (Figure 3 and Table 1).

In the TBARS assay, all tested concentrations of (PhSe)_2_ or Eb induced oxidation of the glycerophospholipid phosphatidylcholine when compared to the control or FeSO_4_, which was used as the pro-oxidant control (Figure 4A,B, respectively). The results of co-treatments with FeSO_4_ also showed that (PhSe)_2_ and Eb, at the tested concentrations, did not prevent iron-induced lipid peroxidation (Figure 4A,B, respectively).

### 3.2. Viral Replication in Calu-3 Cells Exposed to (PhSe)_2_ and Eb

Human type II pneumocytes, target cells of the COVID-19 infectious agent, were infected and treated under different experimental conditions. In an infection model with MOI 0.01, levels ≥ 95% inhibition of SARS-CoV-2 replication were observed after treatment at a 12.5 µM concentration, regardless of the time proposed in this analysis (Figure 5A,B). Importantly, 86% of viral inhibition was also observed after 24 h of treatment with both organoselenium compounds in Calu-3 cells (MOI 0.1) (Figure 5C). Keeping the same virus/cell ratio, reductions of 74% and 81% in the replicative capacity of SARS-CoV-2 were also demonstrated after 48 h of treatment with (PhSe)_2_ and Eb, respectively (Figure 5D). In addition, it was observed that the EC_50_ values of (PhSe)_2_ and Eb in Calu-3 cells, previously infected with MOI 0.01, were 3.9 and 3.8 µM (24 h of treatment) and 3.4 and 2.6 µM (48 h of treatment), while in those infected with MOI 0.1, the values were 4.2 and 3.1 µM (24 h of treatment) and 3.5 and 3.9 µM (48 h of treatment), respectively. Despite the different viral particle numbers and treatment times proposed for this cell line, the concentration required to inhibit 50% of viral replication was similar in all conditions assumed in this study. The EC_50_ values of (PhSe)_2_ and Eb were demonstrated in Table 1.

### 3.3. Flow Cytometric Assays

For the flow cytometry assays, we chose the concentration of 2.5 µM to expose the PBMCs for 24 h. The analyzed toxicological parameters included: morphology and cell cycle changes, viability loss, apoptosis, and necrosis. A decreased size and increased granularity preceded DNA fragmentation and may be an early indicator of apoptosis. The results from Figure 6 relative to the morphology of the PBMCs showed that there were no alterations in the size of the PBMCs exposed to both selenium compounds when compared to the control cells (Figure 6B). Likewise, 2.5 µM of (PhSe)_2_ and Eb did not induce changes in the granularity of the cells (Figure 6C).

The viability and indexes of early apoptosis, advanced apoptosis, and necrosis of PBMCs exposed to 2.5 µM of (PhSe)_2_ and Eb were determined using Annexin V and PI reagents (Figure 7). In this assay, we did not find changes in the cell viability of PBMCs exposed to (PhSe)_2_ and Eb (Figure 7B). Exposure to 2.5 µM of (PhSe)_2_ or Eb also did not induce signals of early apoptosis (Figure 7C), advanced apoptosis (Figure 7D), and/or necrosis (Figure 7E) in the cells.

Data from Figure 8 showed that exposure of PBMCs to 2.5 µM of (PhSe)_2_ and Eb also did not change the cell cycle phases, as indicated by the comparison of G0–G1, S, and G2/M ratios in relation to the control cells.

## 4. Discussion

Since the outbreak and spread of COVID-19, research has been carried out to understand the mechanisms underlying the infection for the further development of approaches capable of containing and/or reducing the spread of the virus and its complications. Although scientists around the world are studying SARS-CoV-2 and COVID-19, and much progress has been made in a short period of time with the vaccination development, there is still an urgent need to repurpose drugs against SARS-CoV-2 and the effects of COVID-19, which has a hallmark of an acute inflammatory response that is exacerbated in the elderly and individuals with chronic disorders, such as obesity and diabetes mellitus [30,31].

Here, we repurposed the use of two organic selenium compounds that have already had their toxicological and pharmacological properties extensively studied by our research group: ebselen (Eb) and diphenyl diselenide ((PhSe)_2_) [24,32,33,34,35,36]. We found very satisfactory and promising results, which showed the effectiveness of both compounds in significantly inhibiting SARS-CoV-2 replication in Calu-3 cells (that recapitulate type II pneumocytes), exhibiting EC_50_ values at concentration ranges that were safe for human cell lines: alveolar lung cells and lymphocytes. Our research group has already demonstrated the activity of (PhSe)_2_ against the SARS-CoV-2 main protease (Mpro) and its replication in Vero E6 cells [36]. In addition, other groups have already demonstrated the activity of Eb in this same enzyme through in silico and in vitro approaches [37,38,39,40,41]. The protease Mpro is highly conserved across the different SARS-CoV-2 “variants of concern” (VOCs) identified, for example, Alpha, Beta, Gamma, Delta, and Omicron [42,43,44]. Therefore, although we used a SARS-CoV-2 B.1 lineage isolate, it is strongly suspected that Eb and (PhSe)_2_ would have similar effects on other SARS-CoV-2 variants. Vero E6 and Calu-3 cell types have particularities in the virus entry process, because the interactions of SARS-CoV-2 Spike proteins and cell receptors are different [45,46,47,48]. Using specially cultured lymphocytes in a battery of the toxicological tests, we found that Eb and (PhSe)_2_, even at concentrations higher than those that inhibited the viral replication, did not induce a loss of viability, apoptosis, or changes in the cell cycle and cell morphology. Indeed, the compounds did not present an effective pro-oxidant effect, as evaluated by a lipid peroxidation estimation.

Selenium (Se) is a trace element, essential for numerous cellular functions in animals. In humans, Se is necessary for the synthesis of at least 25 selenoproteins, where the antioxidant enzymes such as glutathione peroxidase (GPx) and thioredoxin reductase (TrxR) isoforms are highlighted. Of pharmacological importance, same organic selenium compounds, including (PhSe)_2_ and Eb, can mimic GPx and/or be substrates for TrxR [34,49]. Along with the antioxidant activity, Eb and (PhSe)_2_ have pharmacological implications against several experimental human pathologies via anti-inflammatory, anticarcinogenic, antidiabetogenic, and neuroprotective action. It is worth noting here that clinical trials have already been carried out with Eb to treat brain ischemia, noise-induced hearing loss, and bipolar disorder [34,49,50,51].

On the other hand, high levels of Se can be toxic to living organisms. There is strong evidence that many inorganic/organic selenium compounds can oxidize low- and high-molecular-weight thiols, causing redox unbalance [52]. In this sense, the first toxicological works carried out by our research group toward organoselenium compounds demonstrated the pro-oxidant effect of several selenium-containing molecules, including (PhSe)_2_ and Eb. These compounds can inhibit sulfhydryl-containing enzymes such as α-ALA-D and Na^−^K^+^ in both in vitro and in vivo models [53,54,55]. More recently, our works have suggested that the compounds can act as weak electrophiles, a property that also seems to be involved in their antioxidant potential. In fact, by oxidizing -SH groups of keap-1, Eb and (PhSe)_2_ promote the activation of the transcription factor NRF2, which triggers the antioxidant machinery of the cells [34,49,50,51]. The oxidation of Cys from critical enzymes has also been recognized as the main mechanism involved in the antimicrobial activity of the compounds [34,49,50,51]. There is evidence that Eb and (PhSe)_2_ can oxidize critical thiol-containing proteins from viruses, bacteria, and fungi [15]. In the virus context, Eb has already been indicated as a potent inhibitor of key thiol-containing enzymes of the human immunodeficiency virus type 1 (HIV-1) and hepatitis C virus [15]. Regarding (PhSe)_2_, the literature reports its antiviral action against in vitro herpes simplex virus 2 (HSV-2), and in reducing the inflammation in HSV-2-infected mice [56]. (PhSe)_2_ was also efficient against bovine alphaherpesvirus 2, both in vitro and in a sheep model [56].

Especially on SARS-CoV-2, Eb was one of the first molecules to be identified by virtual drug screening as a strong inhibitor of Mpro activity, where the compound displayed inhibition of Mpro activity with an IC_50_ value of 0.67 μM, and viral replication in Vero cells with an EC_50_ value of 4.67 μM [17,18]. Recently, Eb’s efficacy was also observed on SARS-CoV-2 replication in Calu-3 cells, displaying an EC_50_ value of 5 μM, similar to our results [56]. Then, Eb was used in this study as a control for (PhSe)_2_ assays. Molecular and atomistic simulations found that Eb covalently binds to the Cys145 residue from the Mpro active site [35,37]. There is evidence that Eb is also an irreversible inhibitor of PLpro from SARS-CoV-2, with an IC_50_ value of approximately 2 µM [57]. For this enzyme, molecular modeling showed an interaction of the compound with the residue Cys111 from the active site [40]. Herein, our results reinforced the promising findings previously found for Eb, as well as demonstrated its efficacy in human cell lines targeted by SARS-CoV-2 and COVID-19 [22]. Of toxicological significance, the EC_50_ values of compounds on infected Calu-3 cells were very low, reaching concentrations that did not compromise the viability of human cell lines, in the case of Calu-3 and PBMCs. In addition, Eb did not induce deeper signs of cell damage, such as viability loss, morphological changes, lipid peroxidation, and cell death.

Our findings also shed light on the hopeful role of (PhSe)_2_, which exhibited a similar profile of antiviral behavior as Eb, at concentrations that did not cause apparent toxicity to human cell lines. In this sense, our work presented a new promising molecule for studying SARS-CoV-2 inhibition and COVID-19 complications. In terms of the action mechanism, a recent in silico study on the interaction between Mpro and PLpro from SARS-CoV-2 and selenium compounds compared the effects of (PhSe)_2_, Eb, and some derivatives. Although Eb showed higher negative binding energies on the enzyme active sites (Mpro: −6.5 kcal·mol^−1^, Plpro: −6.1 kcal·mol^−1^), (PhSe)_2_ exhibited an interesting effect, with a ΔG value of −6.2 kcal·mol^−1^ for Mpro and −4.1 kcal·mol^−1^ for PLpro [35].

Our research group has already demonstrated the activity of (PhSe)_2_ against the SARS-CoV-2 main protease (Mpro) and its replication in Vero E6 cells [36]. In addition, other groups have already demonstrated the activity of Eb in this same enzyme through in silico and in vitro approaches [37,38,39,40,41]. The protease Mpro is highly conserved across the different SARS-CoV-2 “variants of concern” (VOCs) identified, for example, Alpha, Beta, Gamma, Delta, and Omicron [42,43,44]. Therefore, although we used a SARS-CoV-2 B.1 lineage (Alpha) isolate, it is strongly suspected that Eb and (PhSe)_2_ would have similar effects on other SARS-CoV-2 variants.

We emphasize here that in addition to viral replication inhibition via thiol oxidation, Eb and (PhSe)_2_ have been reported as potent anti-inflammatory agents, an effect that could simultaneously help in the inflammatory phase of COVID-19. In fact, Eb and (PhSe)_2_ have been shown to be efficient in controlling the inflammation in a variety of in vitro and in vivo injury models, including ischemia and reperfusion cerebral, arthritis, food paw edema, pleurisy, and ulcerative colitis models [52]. Reduction of neutrophil infiltrate, inhibition of pro-inflammatory enzymes, reduction of nitric oxide (NO) and pro-inflammatory cytokines’ production, and modulation of the purinergic system are among the phenomena underlying the anti-inflammatory potential of the compounds [15].

Studies have demonstrated that the Se status is low in blood samples from COVID-19 patients and that the in situ SARS-CoV-2 infection significantly reduces the expression of a number of selenoproteins, including SEL P, S, K, GPx, and Trx [58,59]. Se deficiency is also known to be prevalent during other viral infections, including in hepatitis and HIV [60,61]. Although several trials have already reported the beneficial effects of Se supplementation, cellular and molecular findings are still lacking to clarify the role of Se in the viral infection. However, clinical evidence from HIV infection has highlighted that Se supplementation can delay the CD4 decline in HIV-infected individuals, as well as affect the expression of selenoproteins in HIV-infected T cells in a hierarchical manner, with GPx1, GPx4, SEL H, SEL K, SEL S, and SEL T being the mostly sensitive selenoproteins [62,63,64]. In general, these events strengthen the additional benefit that the supplemental compounds containing selenium could have under viral infections. Especially regarding (PhSe)_2_, we have already demonstrated the chronic dietary increases in the levels of Se in the brain, along with the relative transcriptional levels of GPx isoforms and SEL P [65,66], effects that could be interesting under selenium-deficiency conditions.

## 5. Conclusions

Our findings showed the potential effects of Eb and (PhSe)_2_ against SARS-CoV-2 replication at concentrations non-toxic to human cells. Although more efforts need to be made to elucidate the molecular and cellular action mechanisms of compounds on viral replication, we believe that one probable action is via inhibition of the viral proteases Mpro and PLpro, as previously indicated by molecular simulations. Therefore, our promising results in infected cells led us to suggest the use of these molecules in other trials, such as modulation of inflammation prompted by SARS-CoV-2 infection.

## Figures and Tables

**Figure 1 vaccines-11-01222-f001:**
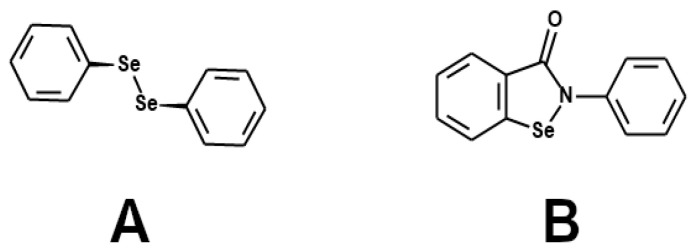
Chemical structures of diphenyl diselenide ((PhSe)_2_) (**A**) and ebselen (Eb) (**B**).

**Figure 2 vaccines-11-01222-f002:**
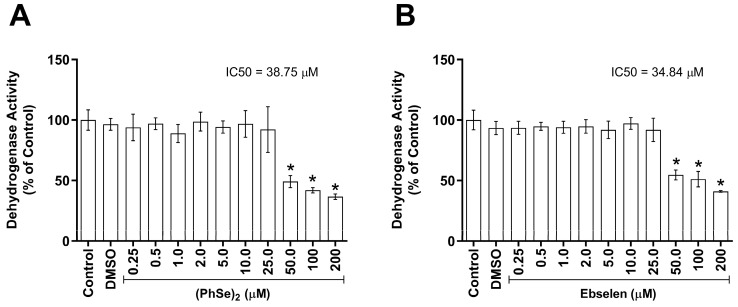
Cell viability of human PBMCs exposed to Eb and (PhSe)_2_. Here, 0.5 × 10^6^ human cells were exposed to (PhSe)_2_ (**A**) or Eb (**B**) at 0.25, 0.5, 1, 2, 5, 10, 25, 50, 100, and 200 μM, for 24 h, at 37 °C in a sterile environment containing 5% CO_2_. Data represent the mean ± SEM of four independent experiments and were analyzed by one-way ANOVA followed by Tukey’s multiple test. * Indicates significant difference when compared to the control (*p* ≤ 0.05).

**Figure 3 vaccines-11-01222-f003:**
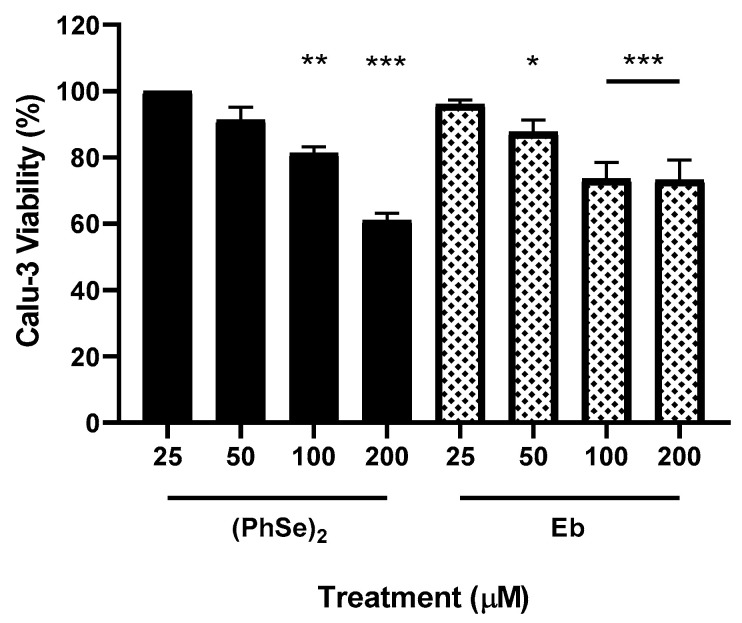
Effect of Eb and (PhSe)_2_ on the viability of non-infected Calu-3 cells. Cells were exposed to Eb or (PhSe)_2_ at concentrations from 25 to 200 µM, for 72 h, at 37 °C. Then, the viability was estimated by the MTT assay and the CC_50_ values of Calu-3 were calculated from the concentration curves. Data represent the mean ± SD of three independent experiments, analyzed by one-way ANOVA, followed by Dunnett’s multiple tests (*n* = 5). * *p* ≤ 0.05, ** *p* ≤ 0.01, and *** *p* ≤ 0.001.

**Figure 4 vaccines-11-01222-f004:**
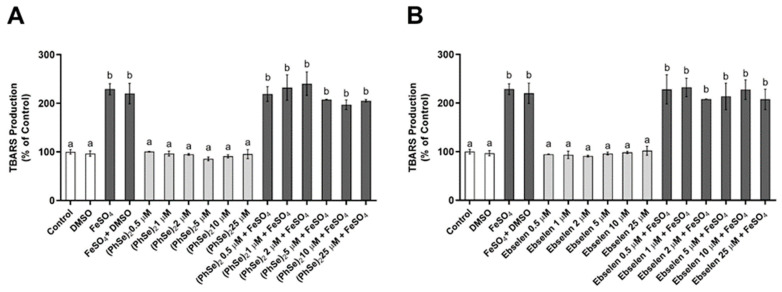
Effects of Eb and (PhSe)_2_ on lipid peroxidation by thiobarbituric acid-reactive substances (TBARS). Graphs illustrate TBARS production by 4.26 mg/mL of phosphatidylcholine exposed, or not, to Eb and (PhSe)_2_ in the presence or absence of 214 µM of FeSO_4_, as previously described. (**A**) (PhSe)_2_ and (**B**) Eb at 0.5, 1, 2, 5, 10, and 25 µM. Data represent the mean ± SEM of four independent experiments, analyzed by one-way ANOVA, followed by Tukey’s multiple tests (*n* = 3). Letters “a” and “b” indicate statistical equivalence.

**Figure 5 vaccines-11-01222-f005:**
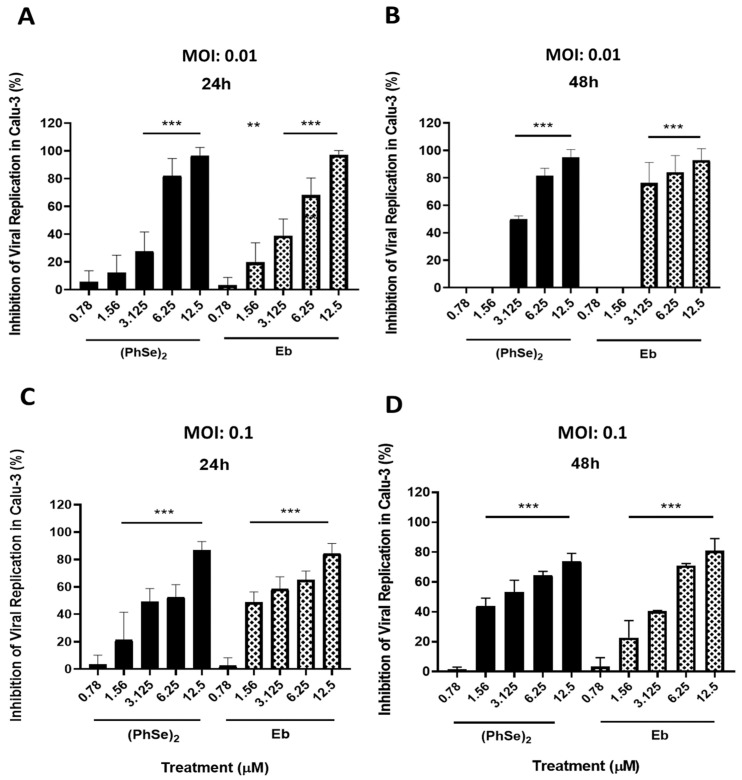
Effect of (PhSe)_2_ and Eb on SARS-CoV-2 replication in Calu-3 cells. Percentage of SARS-CoV-2 replication in Calu-3 cells exposed to Eb or (PhSe)_2_ under different experimental conditions: MOI 0.01 (**A**,**B**) or MOI 0.1 (**C**,**D**), and 24 h (**A**,**C**) or 48 h (**B**,**D**) of treatment time. Statistical analysis was obtained in comparison to the infected control group. Data represent the mean ± SD of five independent experiments, analyzed by one-way ANOVA, followed by Dunnett’s multiple tests. ** *p* ≤ 0.01 and *** *p* ≤ 0.001.

**Figure 6 vaccines-11-01222-f006:**
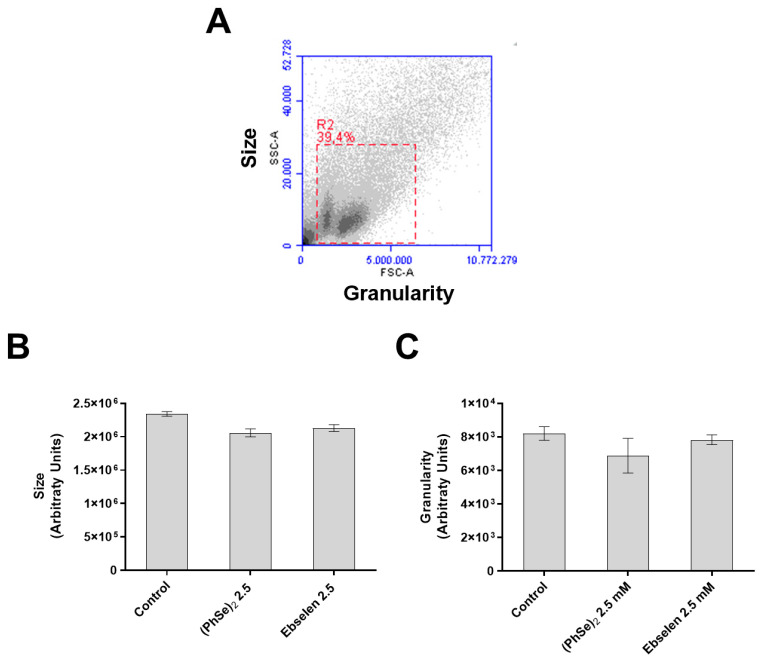
Size and granularity of human lymphocytes exposed to Eb and (PhSe)_2_. Human PBMCs (0.5 × 10^6^ cells/mL per sample) were exposed to (PhSe)_2_ or Eb, at 2.5 μM for 24 h at 37 °C in a sterile environment containing 5% CO_2_. (**A**) Representative histogram of PBMCs by flow cytometry. FSC and SSC axes correspond to the size and granularity of the cells, respectively. (**B**,**C**) Arbitrary units of the size and granularity of the cells. Results represent the mean ± SEM of four independent experiments. Here, 50,000 events were acquired for each sample analysis. The one-way ANOVA followed by Tukey’s multiple test was done. *n* = 6 for each group.

**Figure 7 vaccines-11-01222-f007:**
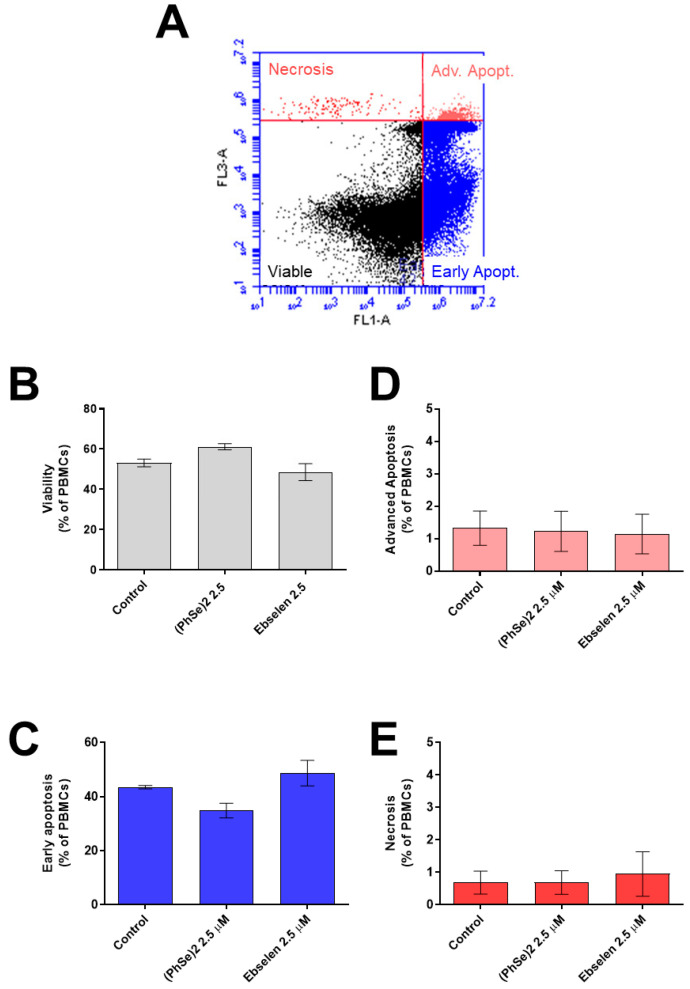
Viability, early apoptosis, advanced apoptosis, and necrosis in lymphocytes exposed to Eb and (PhSe)_2_. Human PBMCs (0.5 × 10^6^ cells/mL per sample) were exposed to (PhSe)_2_ or Eb, at 2.5 μM for 24 at 37 °C in a sterile environment containing 5% CO_2_. Then, the cells were treated with Annexin V/PI reagents, and the fluorescence was measured on a BD Accuri^TM^ C6 flow cytometer. (**A**) Histogram representative of cell staining with Annexin V/PI. Viable cells are in the bottom left quadrant, nonviable necrotic cells are in the top left quadrant, nonviable advanced apoptotic cells are in the top right quadrant, and early apoptotic cells are in the bottom right quadrant. Viability (**B**), early apoptosis (**C**), advanced apoptosis (**D**), and necrosis (**E**), respectively. The results represent the mean ± SEM of three to four independent experiments and are expressed as % of the control. Here, 100,000 events were recorded per sample. No difference was observed between the groups by one-way ANOVA followed by Tukey’s multiple tests.

**Figure 8 vaccines-11-01222-f008:**
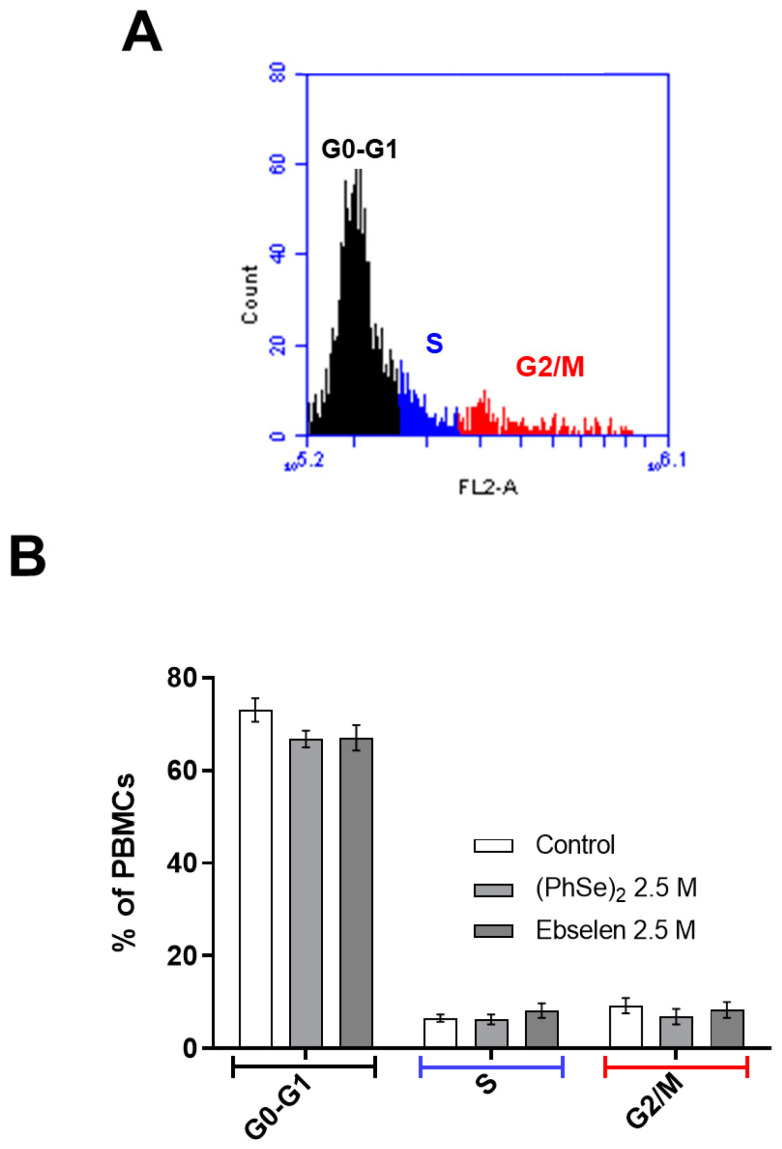
Cell cycle phases of human lymphocytes exposed to Eb and (PhSe)_2_. Cell cycle distribution of PBMCs exposed to (PhSe)_2_ or Eb at 2.5 µM. (**A**) Representative histogram showing cell numbers present in the different phases (G0–G1, S, or G2/M). (**B**) Percentage of resting PBMCs in G0–G1 phases, or S or G2/M phases (proliferative phases). Results were analyzed by one-way ANOVA followed by Tukey’s multiple test and represent the mean ± SEM of six independent experiments.

**Table 1 vaccines-11-01222-t001:** EC_50_ (µM), CC_50_ (µM), and SI of Eb and (PhSe)_2_ in Calu-3 cells.

		MOI 0.01				MOI 0.1			
		24 hpi		48 hpi		24 hpi		48 hpi	
Molecules	CC_50_	EC_50_	SI	EC_50_	SI	EC_50_	SI	EC_50_	SI
(PhSe)_2_	>200	3.9 ± 0.4	51.28	3.4 ± 0.3	58.82	4.2 ± 0.7	47.62	3.5 ± 1.0	57.14
Eb	>200	3.8 ± 0.5	52.63	2.6 ± 0.5	76.92	3.1 ± 0.6	64.52	3.9 ± 0.6	51.28

CC_50_: Drug concentration that promotes the death of 50% of treated cells. EC_50_: Drug concentration capable of reducing the viral effect in 50% of cells. SI: Selectivity index (calculated by the ratio of CC_50_ and EC_50_ values).

## Data Availability

Not applicable.

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
