# Peer review of "Ebselen and Diphenyl Diselenide Inhibit SARS-CoV-2 Replication at Non-Toxic Concentrations to Human Cell Lines"

_vaccines, 2023, doi:10.3390/vaccines11071222_

Round 1

Reviewer 1 Report

In this manuscript, the authors characterized the antiviral activity of two compounds, Ebselen and diphenyl diselenide, against SARS-CoV-2 in calu-3 cells. They evaluated in detail the possible toxic effects of these compounds in different cell lines using several methods.

The manuscript is very well written, and the experiments as well as the results are clearly described. The novelty, however, is not too high, considering that several articles described the antiviral effect of the compounds. Moreover, there is a published article (Huff et al., 2022) reporting the IC50 of Ebselen and analogs against SARS-CoV-2 in Calu3 cells, which authors do not cite in the references.

Huff S, Kummetha IR, Tiwari SK, Huante MB, Clark AE, Wang S, Bray W, Smith D, Carlin AF, Endsley M, Rana TM. Discovery and Mechanism of SARS-CoV-2 Main Protease Inhibitors. J Med Chem. 2022 Feb 24;65(4):2866-2879. doi: 10.1021/acs.jmedchem.1c00566. Epub 2021 Sep 27. PMID: 34570513; PMCID: PMC8491550.

The inhibition mechanism also seems to be described before. Other non-listed literature:

Amporndanai, K., Meng, X., Shang, W. et al. Inhibition mechanism of SARS-CoV-2 main protease by ebselen and its derivatives. Nat Commun 12, 3061 (2021). https://doi.org/10.1038/s41467-021-23313-7

Sahoo P, Lenka DR, Batabyal M, Pain PK, Kumar S, Manna D, Kumar A. Detailed Insights into the Inhibitory Mechanism of New Ebselen Derivatives against Main Protease (Mpro) of Severe Acute Respiratory Syndrome Coronavirus-2 (SARS-CoV-2). ACS Pharmacol Transl Sci. 2022 Dec 23;6(1):171-180. doi: 10.1021/acsptsci.2c00203. PMID: 36650888; PMCID: PMC9797022.  

Menéndez CA, Byléhn F, Perez-Lemus GR, Alvarado W, de Pablo JJ. Molecular characterization of ebselen binding activity to SARS-CoV-2 main protease. Sci Adv. 2020 Sep 11;6(37):eabd0345. doi: 10.1126/sciadv.abd0345. PMID: 32917717; PMCID: PMC7486088.  

Qiao Z, Wei N, Jin L, Zhang H, Luo J, Zhang Y, Wang K. The Mpro structure-based modifications of ebselen derivatives for improved antiviral activity against SARS-CoV-2 virus. Bioorg Chem. 2021 Dec;117:105455. doi: 10.1016/j.bioorg.2021.105455. Epub 2021 Oct 30. PMID: 34740055; PMCID: PMC8556866.  

Minor concerns:

Figure 2. LC50 is not defined.

Figure 3. "foi" = was?

Table 1. "Concentration of molecules" might be change to "Drug concentration". EC50 refers to effect while IC50 refers to inhibition.

Line 264. New coronavirus is confusing, maybe

Line 352. I wouldn´t say that 3-4 uM is a very low EC50.

Line 446. "To bet on" doesn´t seem the most appropriate expression.

Minor editing of English language required

Reviewer 2 Report

Although multiple SARS-CoV-2 vaccines have been highly effective in bringing the pandemic under control, there remains a need for efficacious antiviral therapies to treat less severe infections before they can cause serious illness in non-hospitalized patients.  Although only remdesivir is mentioned in this vein in the article, Paxlovid is now the most frequently prescribed antiviral for Covid-19.  Still, any new addition to our pharmacological arsenal to treat the disease would be a welcomed addition.  Indeed, investigators are continually vetting repurposed drugs that have been shown to be effective against other RNA viruses for their activity against SARS-CoV-2. 

Selenium-based treatments have previously been shown to be effective against viruses such as HIV-1, HSV-2 and hepatitis C.  In this study, two organic selenium compounds, Ebselen and Diphenyl Diselenide, are evaluated for their efficacy against the virus.  The ability to prevent SARS-COV-2 virus replication in the Calu-3 human pneumocyte cell model was evaluated, as well as their potential toxicity to PBMCs, as determined by loss of viability, apoptosis, morphological changes and cell cycle disruption.  The data are convincing that both compounds strongly inhibit virus replication.  The efficacy of these compounds against SARS-CoV-2 notwithstanding, one obvious concern, as with any antiviral, is the potential for toxicity.  The authors have done a good job of addressing this issue.  At concentrations that strongly prevent infection, the compounds did not induce any signs of cellular toxicity, boding well for their ongoing evaluation as SARS-COV-2 antivirals.

This study constitutes a strong initial step in the evaluation of these compounds that justifies further investigation.  There is some potential here.  The authors have been diligent in evaluating both the efficacy and safety of these compounds.  They have also admirably cited all of the positives, as well as the potential negatives, associated with them.

There are only a few minor points that should be addressed in a revision, First, remdesivir is cited as the only FDA-approved therapeutic for SARS-CoV-2.  But, although not FDA-approved, except for high-risk adults, Paxlovid has over the past few years become the antiviral of choice for even moderate cases of the disease.  This should be mentioned in the manuscript.  Second, all the experiments in this study use a specific B.1 lineage isolate of the virus.  Although it is strongly suspected that the compounds would have similar effects on other virus isolates, such as Omicron, this issue should at least be addressed by the authors somewhere in the manuscript.

The use of the English language is rather poor and requires editorial attention.

Reviewer 3 Report

    • Major comments: 

In this study, Guilherme Wildner et al. found that Ebselen and diphenyl diselenide could inhibit SARS-CoV-2 replication in Calu-3 cells, with the EC50 values for Eb and (PhSe)2 after 24 hours post-infection (hpi) were 3.8 μM and 3.9 μM, respectively, and after 48 hpi 2.6 μM and 3.4 μM. 200 μM. The concentration rates tested on viral replication were not toxic to human PBMCs.

      • General concept comments

Here are some considerations for the study:

1.     It should be tested whether Ebselen and diphenyl diselenide can inhibit the main protease activities of SARS-CoV-2 in vitro.

2.     Besides, it should be tested whether the inhibitory effects of Ebselen and diphenyl diselenide were caused by the direct binding of these compounds to the SARS-CoV-2 virion itself.

3.     Do Ebselen and diphenyl diselenide have any prophylaxis effects against SARS-CoV-2?

4.     In Figures 6, 7, and 8, I would suggest also performing an unpaired T-Test of each compound group when compared to the control group.

5.     For Figures 6-8, what treatments were used for the control group?

      • Specific comments:

1)    Line 113, typo of the numbers here.

2)    Lines 178, 192, 205, “100.000” is confusing. Does it mean 100, 000?

3)    In flow cytometry, the FL3 channel was used for PI in 2.7. Viability, apoptosis, and necrosis indexes, while the FL2 channel was used for PI in 2.8. Cell cycle assay, why?

4)    Lines 236-237, control samples were treated with 100% DMSO?

5)    In Figure 6B, it seems like (PhSe)2 treatment group had significant cell changes in size?

Fine.

Round 2

Reviewer 3 Report

The manuscript has been improved, and the authors have addressed most of my concerns. However, experimental data should be provided to test whether Ebselen and diphenyl diselenide can inhibit SARS-CoV-2 Omicron variants’ entries since the Omicron variants are dominant strains nowadays.

Fine.

Author Response

Thank you so much for your quick review.

In this manuscript, we are focusing mainly on Eb and (PhSe)2 anti-SARS-CoV-2 activity in Calu-3 and analyzed their toxic effects on PBMCs. Many published studies with Eb and our article with (PhSe)2 focused on evaluating the mechanism of action against virus protease (Mpro). However, it is possible that these molecules may also act in the initial stages of viral replication. I fact, we realize that the inhibition of the viral replication can be mediated by a direct interaction of the compounds with the virion (possibly some -SH containing protein). We are in the process of purchasing commercial kits to evaluate the interaction of Eb and (PhSe)2 on the native viral spike and variants (including Omicron). Unfortunately, the delivery time for the kits in Brazil is 45 working days, which made it impossible to include these results in this manuscript. These assays with the viral spike and others looking for effect on the initial stages of virus-host cell interaction will compose a new manuscript focused on the evaluation of this mechanism of action in a near future.

Round 3

Reviewer 3 Report

I have no further comments.

Fine.